# High Dietary Cation and Anion Difference and High-Dose Ascorbic Acid Modify Acid–Base and Antioxidant Balance in Dairy Goats Fed under Tropical Conditions

**DOI:** 10.3390/ani13060970

**Published:** 2023-03-07

**Authors:** Sapon Semsirmboon, Dang Khoa Do Nguyen, Narongsak Chaiyabutr, Sutthasinee Poonyachoti, Thomas A. Lutz, Sumpun Thammacharoen

**Affiliations:** 1Department of Physiology, Faculty of Veterinary Science, Chulalongkorn University, Pathumwan, Bangkok 10330, Thailand; 2The Academy of Science, The Royal Society of Thailand, Dusit, Bangkok 10300, Thailand; 3Queen Saovabha Memorial Institute, The Thai Red Cross Society, Pathumwan, Bangkok 10330, Thailand; 4Institute of Veterinary Physiology, Vetsuisse Faculty, University of Zurich, 8057 Zurich, Switzerland

**Keywords:** acid-base, DCAD, high ambient temperature, reactive oxygen, Saanen

## Abstract

**Simple Summary:**

The high environmental temperature of the tropical area compromises milk production in dairy animals, and an increase in the energy required for heat dissipation produces both acid and reactive oxygen species. This response causes an acid–base imbalance and oxidative stress which indirectly affect milk synthesis. We previously showed that a high dietary cation and anion difference (DCAD) supplement increased heat dissipation. Moreover, a high-dose vitamin C (vit C) supplement has been used in dairy ruminants to decrease oxidative stress. The present experiment aimed to investigate the relationship between acid–base balance, oxidative stress, and mammary gland function under high environmental temperature conditions. The results showed that high DCAD and vit C could change acid–base balance and oxidative status. In addition, milk compositions were changed when the oxidative stress was compromised. We conclude that the high environmental temperature compromises milk synthesis in part by changing acid–base balance and oxidative stress.

**Abstract:**

High ambient temperature (HTa) causes acid–base imbalance and systemic oxidative stress, and this may indirectly affect the mammary gland. Furthermore, HTa induces intracellular oxidative stress, which has been proposed to affect cell metabolism directly. We previously showed in dairy goats that the negative effect of HTa was compromised by enhancing heat dissipation during a high dietary cation and anion difference (DCAD) regimen. Moreover, high-dose vitamin C or ascorbic acid (AA) supplements have been used to manage oxidative stress in ruminants. The present study hypothesized that high DCAD and AA supplements that could alleviate the HTa effect would influence the milk synthesis pathway and mammary gland function. The results showed that goats fed with high DCAD had higher blood pH than control goats in the 4th week. The high dose of AA supplement decreases urine pH in the 8th week. The percent reduction of urine pH from the AA supplement was significant in the DCAD group. The high-dose AA supplement decreased plasma glutathione peroxidase activity and malonaldehyde. This effect was enhanced by a high DCAD supplement. In addition, supplementation with AA increased milk protein and citrate and decreased milk FFA. These alterations indicate the intracellular biochemical pathway of energy metabolism and milk synthesis. It can be concluded that a high DCAD regimen and AA supplement in dairy goats fed under HTa could influence the milk synthesis pathway. The evidence suggests that HTa decreases mammary gland function by modification of acid–base homeostasis and oxidative stress.

## 1. Introduction

Milk synthesis is one of the main functions of the mammary gland, and it is compromised by high ambient temperature (HTa). In tropical conditions, milk production of goats and cows in the summer season is lower than production in the winter season [1,2]. This HTa effect can be observed with the activation of evaporative heat dissipation by panting and antioxidant defense [2]. The excess of both responses can lead to acid–base imbalance and oxidative stress [3,4,5] which may indirectly alter mammary gland function. However, research on mammalian cells has shown that HTa could directly affect cell metabolism, and oxidative stress has been proposed to mediate this effect [6].

The typical condition of the tropical climate is year-round HTa and high relative humidity (RH) [1,7]. It is well known that HTa can decrease the dry matter intake (DMI) and milk yield (MY) of lactating dairy goats, while water intake (WI) increases to serve heat dissipation [2,8]. We have demonstrated in goats fed under HTa that evaporative heat dissipation by panting is included in the adaptive mechanism [2]. This panting was observed with a decreased blood buffer and antioxidant capacity [5,9]. During the daytime, the natural HTa increased rectal temperatures (Tr), respiratory rate (RR), and plasma cortisol [10]. The excess HTa response could induce acid–base imbalance and cause respiratory hypocapnia in goats [3,4,5,9]. Blood bicarbonate (HCO_3_) was decreased to compensate for this respiratory acid–base imbalance [11]. During lactation, the rate of metabolism was increased, especially during the early stage. This high rate of metabolism could be the fundamental factor that increases reactive oxygen species (ROS) and acid [12,13]. The excess ROS and acid in the mammary cells are balanced by intracellular and systemic defense systems, including antioxidant defense and buffer systems. Indeed, it has been reported that there is a depletion of antioxidants and buffers during the early stages of lactation [14,15,16]. These depletions were previously reported with compromised mammary gland function [17,18]. Moreover, plasma antioxidants, including glutathione peroxidase (GPx) activity and ascorbic acid (AA), decreased while plasma malonaldehyde (MDA) increased in dairy cows fed under HTa [19,20,21]. These observations suggested that during early-lactating dairy ruminants faced oxidative stress and acid–base imbalance, and these responses could be more prominent in HTa. In addition, the HTa effect that could increase cellular ROS has been proposed to influence glucose metabolism [6]. The present experiment focuses on the HTa effect on milk synthesis and its relationship with systemic antioxidant defense and buffer systems. We have demonstrated previously that high dietary cation and anion difference (DCAD) increased apparent water balance [22]. An increased body water could promote evaporative heat dissipation and compromise the effectiveness of HTa [22,23]. A high DCAD regimen and high-dose AA supplement have been used to manage acid–base imbalance and oxidative stress in dairy ruminants, especially in HTa condition [9,24,25,26]. Therefore, it was hypothesized that milk synthesis and mammary gland function may change when the HTa effect is alleviated, and systemic oxidative stress is compromised.

## 2. Materials and Methods

### 2.1. Study Area, Animals, and Meteorological Data

The current experiment was performed according to the guidelines of the Ethical Principles and Guidelines for the Use of Animals for Scientific Purposes of the National Research Council of Thailand and was approved by the Institutional Animal Care and Use Committee (IACUC) in accordance with the university regulations and policies governing the care and use of experimental animals.

The study was carried out on a commercial dairy goat farm in Nan province, Thailand (latitude 18.81772°, longitude 100.77350°). Twelve early-lactating Saanen goats aged 3 years were included in this study, and all goats were at 2nd lactation. After parturition, all goats were fed in individual pens (90 × 200 cm) and were milked in the milking barn twice a day at 06:00 and 14:00. A concentrate was individually fed to goats twice a day after milking. The total amount of concentrate was calculated based on the MY, while water and Pangola hay were individually provided ad libitum. After the first two weeks of postpartum, all goats were randomly assigned to two groups. These groups were the control group (n = 6) and the DCAD group (n = 6), where dairy goats were fed with a control and a high DCAD diet for 8 weeks, respectively. In the 4th and 8th weeks of the experimental period, all goats were supplemented with a vehicle and a high dose of AA (45 mg/kg body weight intravenously) on the last two consecutive days.

Potassium carbonate (K_2_CO_3_) and sodium hydrogen carbonate (NaHCO_3_) were premixed with concentrate to formulate the high DCAD diet, while the concentrate without supplement was the control diet. The composition and DCAD level of each feed, including control, high DCAD, and Pangola hay, are presented in Table 1. During the experimental period, blood gas parameters, urine pH, Tr, RR, DMI, WI, and MY were measured on both the 4th and 8th weeks of the experimental period. The effect of DCAD on other parameters was evaluated in the 8th week of the experimental period.

The meteorological data of the goat barn were measured by a wet–dry bulb. The data, including ambient temperature (Ta, °C) and humidity (RH, %), were recorded at 06:00 and 15:00. Then, Ta and RH were used to calculate the temperature and humidity index (THI), respectively, based on the following equation [27]:THI = (Tdb + Twb) ∗ 0.72) + 40.6(1)
where Tdb = dry bulb temperature and Twb = wet bulb temperature.

### 2.2. Data Collection, Measurement, and Analysis

To measure Tr, a thermometer (digital clinical thermometer C202, Terumo, Tokyo, Japan) was placed against the rectum wall for 1 min. The RR was measured by counting the movement of the chest. Both Tr and RR were measured at 13:00. Milk was individually weighed (8 h, 06:00–14:00) and 50 mL of milk sample was collected for milk composition analysis. The volume of 4% fat-corrected milk (4%FCM; 8 h, 06:00–14:00) was calculated based on the dairy goat formula [28]. To determine feed intake and WI, the offered weight of feed and water was subtracted from the refused weight. The value of DMI was calculated based on the dry matter feed.

The acid–base balance of the dairy goats was evaluated using blood gas parameters and urine pH. To evaluate blood gas, 1 mL of blood was collected from the jugular vein at 15:00 and stored in a 1 mL heparinized syringe with a cap. This syringe was transported with crushed ice and was measured within 30 min after collection using the iSTAT blood gas analyzer (Abbott, Freehold, NJ, USA). In addition, urine was collected via a Foley catheter and was connected with a urine collection bag. This urine bag was filled with liquid paraffin to preserve anaerobic conditions from 06:00 to 15:00. Then, all urine samples were poured into a transport tube and gently mixed before measuring the urine pH using a pH meter (Digicon PH-218, Sangchai Meter Co., Ltd., Bangkok, Thailand).

Each blood sample (4 mL) was collected (15:00) and stored in a lithium-heparin tube. Plasma was separated by centrifugation (12,000× *g* at 4 °C for 10 min; Universal 32 refrigerated benchtop centrifuge; Andreas Hettich GmbH, Tuttlingen, Germany). The plasma was aliquoted and kept under −80 °C for cortisol (CBS-E18048G; CUSABIO, Houston, TX, USA), glutathione peroxidase activity (ab102530; Abcam, Cambridge, UK), and malondialdehyde measurement (ab118970; Abcam, Cambridge, UK).

A milk sample (50 mL) was aliquoted into two plastic bottles and stored at −20 °C until analysis. Milk from the first bottle of the sample was used for milk composition analysis by Milko Scan (FT2; Foss, Hilleroed, Denmark) and the parameters were fat, protein, lactose, citrate, and free fatty acids (FFA). Milk from the second bottle was centrifuged at 300× *g*, 4 °C (Thermo Sorvall Legend XTR refrigerated centrifuge; Thermo Scientific, Osterode am Harz, Germany) for 10 min to obtain skim milk. Then, the skim milk was deproteinized by adding 6 N HCL and centrifuged for 5 min at 10,000× *g*, 4 °C (Thermo Sorvall Legend XTR refrigerated centrifuge; Thermo Scientific, Osterode am Harz, Germany). After discarding the pellet, the sample was neutralized by 6 N NaOH and centrifuged again for 5 min at 10,000× *g* 4 °C. Lastly, the supernatant from this step was used for both glucose (EnzyChromTM Glucose assay kits (EBGL-100); BioAssay System, Hayward, CA, USA) and glucose-6-phosphate (G6P) (EnzyChrom^TM^ Glucose-6-Phosphate assay kit (EG6P-100); BioAssay System, Hayward, CA, USA) measurement.

### 2.3. Statistical Analyses 

All data are presented as mean ± standard error of the mean (SEM). Data containing two factors were analyzed using repeated two-way analysis of variance, and the Bonferroni test was used for post hoc analysis. Statistical significance was declared at a *p*-value of <0.05.

## 3. Results

### 3.1. Ambient Condition and the Effect of DCAD and AA Supplementation on Tr, RR, DMI, WI, and MY

In the 4th week of the experimental period, the average of Ta, RH, and THI during the morning were 26 ± 0.4 °C, 92 ± 0.1%, and 78 ± 0.6, respectively. During the afternoon, these ambient conditions were 35 ± 1.6 °C, 61 ± 1.9, %, and 86 ± 2.4, respectively. The Ta difference between morning and afternoon (Δta) was 9 °C. The ambient conditions on the 8th week of the experimental period during morning and afternoon were 25 ± 0.4 °C, 92 ± 0.1%, 76 ± 0.6 and 33 ± 1.5 °C, 65 ± 4.3%, 84 ± 0.9, respectively. The calculated Δta was 8 °C. There was no effect (*p* > 0.05) from both high DCAD and AA on all physiological and behavioral responses to Hta from both the 4th and 8th weeks of the experimental period, including RR, Tr, DMI, WI, MY, and 4% FCM (Table 2).

### 3.2. Effect of DCAD and AA Supplementation on Blood Gas Parameters, and Urine pH 

The high DCAD regimen influenced systemic acid–base balance in the present experiment. In the 4th week, the high DCAD supplementation significantly affected the blood pH (Table 3). The value of pH from the DCAD/vehicle group was significantly higher than that from the control/vehicle group (*p* < 0.05). Similarly, there was a tendency for DCAD to affect blood HCO_3_ (Table 3). In addition, the high dose of AA treatment did not affect blood gas parameters, but it tended to affect the urine pH (Table 3). 

In the 8th week of the experimental period (Table 3), the effect of high DCAD on the blood gas parameters was no longer observed. However, the high DCAD supplementation tended to affect the urine pH. The urine pH tended to be higher in the DCAD group than in the control group. In contrast, the high-dose AA supplement significantly affected urine pH (Table 3). Interestingly, the percentage reduction in urine pH by AA was more pronounced in the control group in the 8th week of the experimental period (Figure 1).

### 3.3. Effect of DCAD and AA Supplementation on Plasma Antioxidant Capacity and Plasma Cortisol

The evaluation of the effect of DCAD and AA supplementation on systemic antioxidant capacity was performed in the 8th week of the experimental period. The DCAD supplement tended to affect plasma GPx activity (Figure 2a) while having no effect on plasma MDA (Figure 2b). The high dose of AA supplement significantly affected both plasma GPx activity and MDA level (Figure 2a,b, *p* < 0.05). The GPx activity from the control and DCAD groups was decreased by a high-dose AA supplement, while the reduction in MDA was more prominently observed in the DCAD group. In addition, both high DCAD and AA supplementations did not affect the plasma cortisol (Figure 3).

### 3.4. Effect of DCAD and AA Supplementation on Milk Composition and Yield of Composition

There was no effect of high DCAD on major and minor milk compositions and the yield of each composition in this study (Table 4). In contrast, the concentration of milk protein, citrate, and FFA was affected by the high-dose AA supplement (Table 4). The concentration of milk protein and citrate was higher (*p* < 0.05) while FFA concentration was lower (*p* < 0.05) in the high-dose AA-supplemented goats when compared to the vehicle-supplemented goats. The yield of citrate was higher and the yield of FFA was lower with the high-dose AA supplement (*p* < 0.05). In addition, the interaction of both factors tended to affect the yield of milk fat (Table 4). 

## 4. Discussion

The current study demonstrated that, under HTa conditions, the interference of acid–base balance and oxidative status could modify mammary gland function. Milk protein is the only major milk composition that was modified. Milk citrate and FFA were the two other components of minor milk composition that could be modified when systemic antioxidant was spared. 

In the present experiment, the range of afternoon Ta was around 33–35 °C and the ΔTa value was approximately 8–9 °C. These levels of the afternoon Ta and ΔTa conditions were comparable to the HTa of tropical conditions [1,2,3] that could activate physiological mechanisms and affect milk production. This was supported by the responses from RR, Tr, and WI that were in line with our previous results and indicated the activation of heat dissipation [2,3,4]. It appears likely that the responses were sufficient for acid–base imbalance (i.e., sporadic respiratory hypocapnia), depleted antioxidants and decreased mammary gland function in dairy goats and cows [1,2,3]. Importantly, the present experimental conditions and animals were devoid of heat stress because plasma cortisol was kept at a normal level and also close to our previous study, in which the response did not involve the hypothalamic–pituitary axis [3]. Both high DCAD and AA supplementation from the present experiment did not alter all HTa-induced behavioral and physiological responses (Table 1). It should be noted that this outcome may be influenced by the limitation of the present experiment, i.e., the number of goats from each group. Nevertheless, these results suggest, in part, that RR, Tr, DMI, and WI are part of the physiological responses of the low degree of HTa and depend directly on ambient conditions. The conclusion is supported at least by our previous study. First, the low level of HTa could activate a specific hypothalamic neural circuit to decrease food intake without stress activation [29,30]. Second, at a high level of HTa, high DCAD has been shown to activate heat dissipation in dairy goats by increasing WI and body water turnover and subsequently decreasing the rate of Tr increment [22,23]. 

High DCAD regimen and AA supplementation in the present experiment could change the systemic acid–base balance and antioxidant capacity. The current study demonstrated that a high DCAD regimen increased blood pH (and perhaps blood HCO_3_) in the 4th week of the experimental period, but this response was no longer observed in the 8th week of the experimental period. Although the effect of high DCAD on blood gas was not observed in the 8th week of the experimental period, the urine pH tended to be high in the presence of high DCAD. This indicates an adaptive response of bicarbonate load to the systemic body fluid in the 4 consecutive weeks of the experimental period. During the first 4 weeks of the high DCAD regimen, the acid–base balance of intravascular fluid shifted toward the metabolic alkalosis direction. The kidney function that regulates urine pH was activated during the second 4 weeks of the present experiment. This is in contrast to when goats were given a low DCAD regimen [4] and indicated the threshold of kidney function in terms of acid and base regulation. Specifically, a low DCAD regimen or acid load [4] could activate kidney function faster than a high DCAD regimen or bicarbonate load. A high dose of AA supplementation influenced systemic acid–base balance by decreasing urine pH. This effect is straightforward because AA is a urine-acidified agent and is excreted mainly through the urine [31,32]. In addition, the high DCAD regimen could reverse the effect of the AA supplement on urine pH reduction. This would suggest the effect of DCAD and the interaction of both treatments on the kidney control mechanism of acid-base balance. Although high DCAD and AA treatments could modify acid–base homeostasis, all blood gas parameters from the present experiment were kept in a normal range [33].

Dairy ruminants fed under HTa conditions require high energy for both lactation and heat dissipation [34]. To produce a lot of energy, the level of the metabolic rate increases, leading to an increase in ROS [13]. Although both treatments from the present experiment did not change plasma cortisol, a high dose of AA supplementation could interfere with the antioxidant system of the body fluid by decreasing plasma GPx activity without the interference of the DCAD effect. The effect of AA in the present experiment is in line with previous results suggesting that AA treatment could reverse GPx activity in rats under heat stress [35]. The effect of AA and DCAD supplementation on plasma MDA in the present experiment is complicated. The concentration of plasma MDA from AA supplementation was lower than that from the vehicle only from the high DCAD group. This suggested the interference from both AA and DCAD supplementation on oxidative stress production under HTa conditions. In general, plasma MDA seemed to be higher under HTa conditions in dairy goats and cows, i.e., during summer [2,19,36]. It has been shown previously that a DCAD regimen could in part increase the heat dissipation mechanism and alleviate the effect of HTa [22]. We anticipate that a high dose of AA supplementation has the potential to decrease MDA production. Interestingly, the effect of AA might be masked by DCAD based on acid-base balance, but the effect of this supplement on MDA production is pronounced when the heat dissipation mechanism is accelerated by DCAD supplementation. This suggested a synergistic effect of both supplements on oxidative stress. 

Although most of the major and minor milk compositions were not changed by the high DCAD regimen, a high dose of AA supplementation could change the concentration of milk protein, citrate, and FFA. This suggested that milk synthesis and metabolic pathways of mammary glands from the dairy goats fed under HTa conditions could be modified mainly by a high dose of AA supplementation. The effects were associated with the systemic antioxidant effect of AA supplementation. Whether these were the direct AA effect on mammary gland oxidative status or indirect from the AA effect on systemic antioxidants remains obscure and needs further appropriately controlled experiments. A high dose of AA supplementation increased milk protein concentration in the present experiment. This information supports the hypothesis that the HTa conditions of tropical areas could influence milk protein synthesis by oxidative stress modification [10]. We have shown previously that both milk citrate and FFA were under control by breed and type of roughage in dairy cows fed under HTa conditions [37]. In the present experiment, the concentration of milk citrate was higher and the concentration of milk FFA was lower in the high-dose AA supplementation group than those in the control/vehicle group. These results suggested that a high dose of AA could modify intracellular citrate and perhaps triglyceride synthesis in mammary cells. In ruminants, citrate, as well as pentose monophosphate pathway (PPP) [37], plays a role in fatty acid synthesis mainly by providing the reducing equivalents via the isocitrate cycle, and the concentration of milk citrate has been shown to relate to de novo fatty acid synthesis [38]. A decrease in milk citrate during mid-lactation is related to an increase in medium-chain fatty acids that are incorporated into milk triglyceride. The phenomenon has been demonstrated well with the effect of the lactation stage in dairy cows [37,38]. Nicotinamide adenine dinucleotide phosphate (NADPH) in the mammary epithelium is not only the crucial intracellular reducing equivalent for fatty acid synthesis, but the molecule plays an important role in anti-oxidant recycling as well [39]. We argue that the higher citrate concentration from the present experiment came from the high dose of AA supplementation that could restore the NADPH molecule and subsequently spare the intracellular citrate. However, this hypothesis could not explain the higher milk citrate that is associated with the lower milk FFA, as mentioned above [37,38]. Another possible explanation is that the high dose of AA could modify fatty acid incorporation of the triglyceride synthesis pathway, i.e., preventing medium-chain fatty acid incorporation. This would subsequently spare the intracellular citrate as well. Although the present information could not provide a definite conclusion about the mechanism of a high dose of AA supplementation on milk citrate and FFA, it suggests that a high dose of AA supplementation might have a direct effect on mammary gland function. Moreover, it has been hypothesized that the high concentration of intracellular concentration of G6P and the high ratio of G6P and glucose (G6P: G) reflect the high degree of oxidative stress during high milk yield of the early lactation [40]. The comparable value of intracellular glucose, G6P, and the ratio of G6P: G from the present experiment suggested that during the oxidative stress transition, the mammary epithelium cells from dairy goats under HTa conditions favor the isocitrate/citrate cycle rather than PPP. 

## 5. Conclusions

This study showed that a high DCAD regimen and high AA supplement influence acid-base balance under HTa conditions. Furthermore, oxidative stress was compromised mainly by a high-dose AA supplement, and the DCAD supplement enhanced this response. These observations were demonstrated in relation to the alteration of milk protein, citrate, and FFA. Therefore, the information suggests that acid–base balance, systemic, and mammary gland oxidative stress might involve in milk synthesis and mammary gland function under HTa conditions.

## Figures and Tables

**Figure 1 animals-13-00970-f001:**
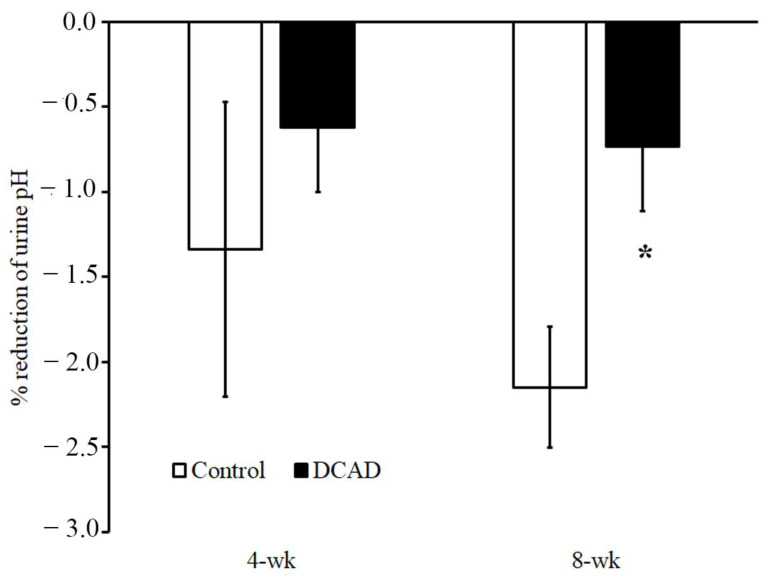
Percentage reduction of urine pH by high-dose AA supplementation from the control vehicle during the 4th and 8th weeks of the experimental period. * Significant effect of DCAD.

**Figure 2 animals-13-00970-f002:**
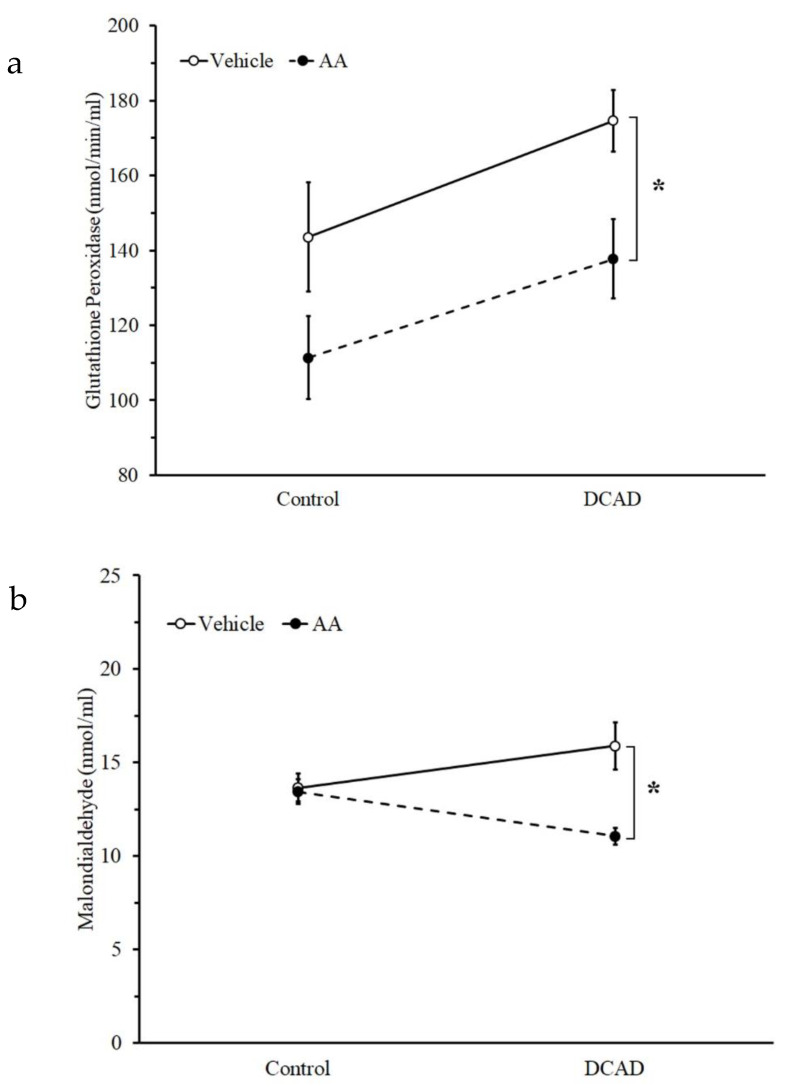
Effect of DCAD and AA supplement on (**a**) plasma glutathione peroxidase (GPx), and (**b**) malonaldehyde (MDA) during the 8th week of the experimental period. * Significant effect of AA.

**Figure 3 animals-13-00970-f003:**
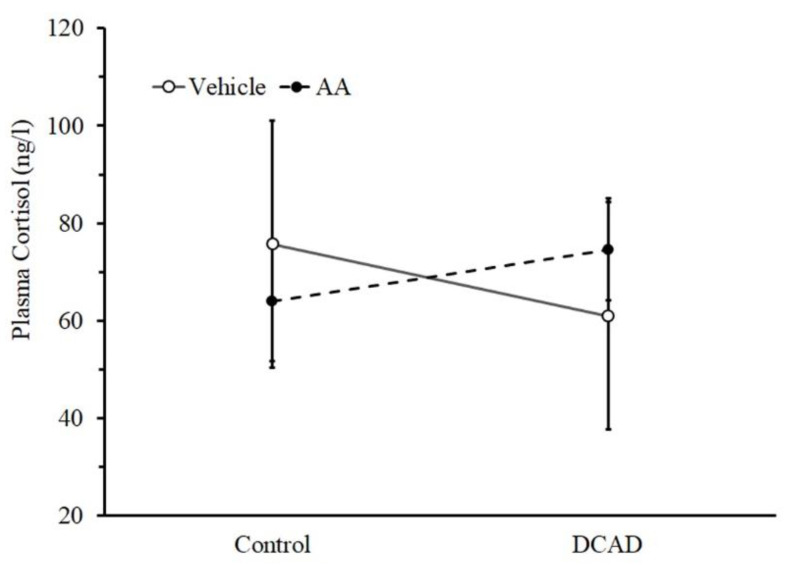
Effect of DCAD and AA supplement on plasma cortisol during the 8th week of the experimental period.

**Table 1 animals-13-00970-t001:** Chemical compositions of the feed component (% on dry matter basis).

Feed Composition (%)	Hay	Control	DCAD
Dry matter	92.68	91.26	87.76
Protein	4.2	16.1	15.5
Crude Fat	1.0	3.9	2.6
NDF	78.9	-	-
ADF	48.4	-	-
Ash	7.7	7.1	7.1
Ca	0.8	1.2	1.4
P	0.1	0.5	0.6
Na	14	1	13
K	36	32	57
Cl	9	15	17
S	12	12	8
DCAD (mEq/100 g DM)	29	6	45

Note: control and DCAD, a concentrate that was used in control and DCAD groups.

**Table 2 animals-13-00970-t002:** Effect of DCAD and AA supplement on behavioral and physiological responses during the 4th and 8th weeks of the experimental period.

	Control		DCAD			*p*-Value		
	Vehicle	AA	Vehicle	AA	SEM	DCAD	AA	DCAD × AA
4th week of supplement								
RR (TPM)	67	90	93	102	21.84	0.28	0.11	0.47
Tr (°C)	38.9	39.0	38.7	38.9	0.30	0.60	0.23	0.95
DMI (%kgBw)	3.04	3.05	3.40	3.58	0.21	0.31	0.35	0.26
WI (%kgBw)	14.2	14.0	13.5	15.2	1.36	0.92	0.19	0.13
MY (kg/8 h)	0.53	0.56	0.53	0.54	0.03	0.91	0.19	0.48
4%FCM (kg/8 h)	1.21	1.28	1.30	1.32	0.08	0.78	0.23	0.51
8th week of supplement								
RR (TPM)	51	60	81	70	18.76	0.22	0.95	0.22
Tr (°C)	38.9	38.9	39.0	38.7	0.20	1.00	0.05	0.18
DMI (%kgBw)	3.22	3.47	3.70	3.49	0.28	0.56	0.83	0.08
WI (%kgBw)	12.6	12.7	12.4	12.0	2.13	0.88	0.85	0.75
MY (kg/8 h)	0.46	0.48	0.48	0.48	0.04	0.90	0.51	0.35
4%FCM (kg/8 h)	1.03	1.08	1.15	1.13	0.08	0.69	0.61	0.32

Abbreviations; RR, respiration rate; Tr, rectal temperature; DMI, dry matter intake; WI, water intake; MY, milk yield; 4%FCM, 4% fat corrected milk.

**Table 3 animals-13-00970-t003:** Effect of DCAD and AA supplement on blood gas parameters during the 4th and 8th weeks of the experimental period.

	Control		DCAD			*p*-Value		
	Vehicle	AA	Vehicle	AA	SEM	DCAD	AA	DCAD × AA
4th week of supplement								
Blood pH	7.43	7.44	7.50	7.49	0.03	0.02	0.69	0.24
PCO_2_ (mmHg)	35.37	33.82	34.42	34.15	2.54	0.86	0.40	0.55
HCO_3_ (mmol/L)	23.35	22.68	27.28	25.50	2.65	0.09	0.28	0.62
Urine pH	8.17	8.06	8.13	8.08	0.1	0.9	0.07	0.47
8th week of supplement								
Blood pH	7.47	7.48	7.48	7.50	0.03	0.46	0.18	0.74
PCO_2_ (mmHg)	35.02	35.00	36.45	35.87	1.57	0.36	0.65	0.67
HCO_3_ (mmol/L)	25.48	26.23	27.32	28.02	1.70	0.31	0.32	0.97
Urine pH	8.22	8.04	8.31	8.24	0.05	0.0547	<0.01	0.025

**Table 4 animals-13-00970-t004:** Effect of DCAD and AA supplement on milk composition and yield of composition.

	Control		DCAD			*p*-Value		
	Vehicle	AA	Vehicle	AA	SEM	DCAD	AA	DCAD × AA
Milk composition								
Fat (%)	4.68	4.73	5.05	4.81	0.27	0.63	0.42	0.21
Protein (%)	2.68	2.74	2.61	2.67	0.05	0.46	0.01	0.89
Lactose (%)	4.25	4.23	4.24	4.26	0.06	0.99	0.99	0.12
Citrate (%)	0.125	0.133	0.098	0.110	0.007	0.28	0.01	0.57
FFA (%)	0.57	0.51	0.88	0.77	0.07	0.15	0.01	0.29
Glucose (µM)	70	72	79	95	14.13	0.61	0.15	0.27
G6P (µM)	166	182	170	186	29.44	0.83	0.21	0.97
G6P: Glu ratio	2.92	3.10	3.73	3.11	0.77	0.72	0.50	0.23
Yield of composition (8-h)								
Fat (g)	20.98	22.42	23.76	22.02	1.96	0.72	0.85	0.07
Protein (g)	12.16	13.11	12.54	12.65	0.92	0.98	0.19	0.29
Lactose (g)	19.34	20.33	20.40	20.26	1.40	0.89	0.47	0.36
Citrate (g)	0.54	0.61	0.50	0.56	0.06	0.65	0.02	0.99
FFA (g)	0.026	0.025	0.039	0.033	0.003	0.18	0.03	0.10
Glucose(µmol)	30	32	39	49	9.20	0.48	0.13	0.32
G6P (µmol)	76	86	84	91	14.30	0.83	0.18	0.73

## Data Availability

The data used to support the findings of this study are available from the corresponding author (ST) upon request.

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
