# Peer review of "High Dietary Cation and Anion Difference and High-Dose Ascorbic Acid Modify Acid–Base and Antioxidant Balance in Dairy Goats Fed under Tropical Conditions"

_animals, 2023, doi:10.3390/ani13060970_

Round 1

Reviewer 1 Report

Authos demonstrated that high DCAD regimen and high AA supplement influence 332 acid-base balance under HTa conditions in goat. In general, the work is well done in terms of methods, analysis and reporting of results. The manuscript is reasonably well done. There are however a few improvements that are required.

1. L100ーL104 Please clarify the number of animals in each group. Also describe in detail.

2. Table 2 What is 4-PP supplement which means 4th week of supplemnt? Please clarify this.

3. Discussion Please add the information the effects about co-treatment of DCAD and AA. DCAD may mask the effect of AA.

Author Response

Dear The editor in chief of Animals and reviewers

       We would like to thank to the editor that provide opportunity and consider our manuscript. We also extremely appreciate for the valuable questions and suggestions made by all our reviewers. Below is our responses point by point and we do hope that our revised manuscript is good enough to publish soon.

Sincerely yours

Sumpun

Reviewer #1

Authos demonstrated that high DCAD regimen and high AA supplement influence 332 acid-base balance under HTa conditions in goat. In general, the work is well done in terms of methods, analysis and reporting of results. The manuscript is reasonably well done. There are however a few improvements that are required.

  1. L100ーL104 Please clarify the number of animals in each group. Also describe in detail.

Ans: We do apologize for unclear description.

            In this study, 12 early-lactating Saanen goats aged 3 years were included and they are at 2nd lactation. After two weeks of post-parturition, they were randomly allocated into two groups, control and DCAD. Each group consisted of 6 lactating dairy goats. The inclusion criteria were already detailed on L95-96. Besides, we added (n=6) to control and DCAD group to declare the number of animals in each group (L102).

L95-96: “ … Twelve early-lactating Saanen goats aged 3 years…”

L102 “… These groups were the control group (n = 6) and the DCAD group (n = 6), ….”

  1. Table 2 What is 4-PP supplement which means 4th week of supplement? Please clarify this.

Ans: Thank you very much for your comment, we apologize that this was not clear.

In table2, we did change this 4-PP and 8-PP to 4th and 8th week of supplement (L175-178)

  1. Discussion Please add the information the effects about co-treatment of DCAD and AA. DCAD may mask the effect of AA.

Ans: Thank you, we do appreciate this suggestion and we have discussed more information.

Even though AA is a week acid, we did not see the effect of this supplement on blood gas. The question is whether this effect was masked by the high DCAD. Without DCAD, many previous studies showed that a high dose AA injection was not enough to affect blood gas parameters. In contrast, the presence of high DCAD might mask the effect of AA based on urine pH. Moreover, the reduction of plasma MDA was more pronounced in the presence of DCAD. This suggested a synergistic effect of both supplements on plasma oxidative stress. Then, we added this discussion on L289-291 and L292-293.

L289-291 “….MDA production. Interestingly, the effect of AA might be masked by DCAD based on acid-base balance, but the effect of this supplement on MDA production …..”

L292-293 “… This suggested a synergistic effect of both supplements on oxidative stress...”

Reviewer 2 Report

The effects of a high DCAD diet and ascorbic acid on blood parameters and the production performance of dairy goats in the tropical environment were studied. The results found that a high DCAD regimen and AA supplement in dairy goats fed under high ambient temperatures could influence the milk synthesis pathway.

1. The author needs to strengthen the potential regulatory role and related mechanism of DCAD diet in heat stress in the Introduction.

2. The author only gave the calculation method of THI, but did not provide the change data of temperature and humidity index of the sheep shed. It is suggested that the author supplement it to judge the degree of heat stress of the experimental animals.

3. The protein yield in Table 4 is lower than the general level. The authors need to discuss this in detail.

Author Response

Dear The editor in chief of Animals and reviewers

       We would like to thank to the editor that provide opportunity and consider our manuscript. We also extremely appreciate for the valuable questions and suggestions made by all our reviewers. Below is our responses point by point and we do hope that our revised manuscript is good enough to publish soon.

Sincerely yours

Sumpun

Reviewer #2

The effects of a high DCAD diet and ascorbic acid on blood parameters and the production performance of dairy goats in the tropical environment were studied. The results found that a high DCAD regimen and AA supplement in dairy goats fed under high ambient temperatures could influence the milk synthesis pathway.

  1. The author needs to strengthen the potential regulatory role and related mechanism of DCAD diet in heat stress in the Introduction.

Ans: Thank you, we do appreciate this suggestion and we also added more information.

From our previous publication, high DCAD diet increased apparent water balance. This increases body water which is responsible for evaporative heat dissipation. Besides, an increased heat dissipation decreased percentage change of Tr during daytime while DMI was increased by this diet. These suggested that this DCAD could promote evaporative heat dissipation and decrease the effect of HTa. Therefore, we added this information on L17-81.

L80-81; “…… dietary cation and anion difference (DCAD) increased apparent water balance [22]. An increased body water could promote evaporative heat dissipation …….. [22,23].”

  1. The author only gave the calculation method of THI, but did not provide the change data of temperature and humidity index of the sheep shed. It is suggested that the author supplement it to judge the degree of heat stress of the experimental animals.

Ans:  This is very important, thank you very much. We addressed information based on Ta, RH, and THI during the morning and afternoon (L166-171). This data set was measured inside the goat shed to show the ambient condition or surrounding environment. Actually, the chronological ambient information was measured and report as ΔTa that was calculated based on the difference between morning to afternoon which we know from our previous experiment that the value represents the highest degree of daily ΔTa.

  1. The protein yield in Table 4 is lower than the general level. The authors need to discuss this in detail.

Ans:  Thank you for suggestion and we did revise our milk protein concentration and yield.

The percentage of milk protein was in line with the previous report, the yield was calculated based on the 8-hour milk yield instead of 24-hour (24 hour). This lower amount of 8-hour milk yield could explain the lower yield in this study. To visualize this information, we did added “ Yield of composition (8-hours)” in table 4.

L224-225 “In table 4, .. Yield of composition (8-hours)…”

Reviewer 3 Report

Introduction

Acceptable

Material and methods

Line 94, why did you use only 12 goats (3 goats per treatment as there are four treatments factorially arranged). Is this adequate replication to detect significant difference?

Line 98-10. After the first two weeks 99 of postpartum, all goats were randomly assigned to two groups. Is the milk yield of the goats were not different? Why they randomly allocated without blocking for their difference in milk yield?

Line 112-113, Present the NDF and ADF composition of the control and DCAD.

Results

Line 169-172, the authors indicated “There was no 169 effect (P>0.05) from both high DCAD and AA on all physiological and behavioral responses to HTa from both the 4th and 8th weeks of the experimental period, including RR, 171 Tr, DMI, WI, MY, and 4% FCM”. Is this not expected taking into account the extremely lower replication (3 per treatment) utilised in this study. The degree of freedom for the error term is 8, which is lower than the recommended value (12) for an error term.

Line 173-175, abbreviations used in Table 2 need to be defined as a footnote as table need to provide adequate information without referring to the other section of the manuscript.

It is generally well described. The main weakness or limitation of this study is that the number of replications utilized is too low (3 replication per treatment).

Discussion

Line 294-295, In a statement “Whether these were the direct AA effect of mammary gland oxidative status” change direct AA effect ‘of’ to direct AA effect ‘on’.

Conclusion

First part is acceptable but the later part seems speculative.

Author Response

Dear The editor in chief of Animals and reviewers

       We would like to thank to the editor that provide opportunity and consider our manuscript. We also extremely appreciate for the valuable questions and suggestions made by all our reviewers. Below is our responses point by point and we do hope that our revised manuscript is good enough to publish soon.

Sincerely yours

Sumpun

Reviewer #3

Introduction

Acceptable

Material and methods

Line 94, why did you use only 12 goats (3 goats per treatment as there are four treatments factorially arranged). Is this adequate replication to detect significant difference?

Ans: We do apologize for our unclear explanation.

In this study, we included 12 lactating goats which were randomly allocated to control group (n=6) and DCAD group (n=6). Besides, both groups were supplemented with vehicle and AA on the last consecutive day. We did add (n=6) on L102 to improve our explanation.

L102 “…. groups were the control group (n = 6) and the DCAD group (n = 6), ….”

Line 98-10. After the first two weeks 99 of postpartum, all goats were randomly assigned to two groups. Is the milk yield of the goats were not different? Why they randomly allocated without blocking for their difference in milk yield?

Ans: Yes, we randomly allocated animals without blocking. The goat originated from the same herd and there was no difference in milk yield between control and DCAD groups.

Line 112-113, Present the NDF and ADF composition of the control and DCAD.

Ans: Thank you very much to point us this unclear description.

In this study, TMR feeding method was not adopted. The concentrate was fed individually after milking while hay was fed all time in each cage. The chemical composition of control and DCAD in table was a chemical composition of a concentrate that used in each group. In general, concentrate was an energy feed which had low NDF and ADF. Then, we added footnote under table 1 on L113 to explain this information.

L114: “… Note: control and DCAD, a concentrate that was …...”

Results

Line 169-172, the authors indicated “There was no 169 effect (P>0.05) from both high DCAD and AA on all physiological and behavioral responses to HTa from both the 4th and 8th weeks of the experimental period, including RR, 171 Tr, DMI, WI, MY, and 4% FCM”. Is this not expected taking into account the extremely lower replication (3 per treatment) utilised in this study. The degree of freedom for the error term is 8, which is lower than the recommended value (12) for an error term.

Ans: We do apologize for our explanation.

In a previous study, the effect of DCAD when compared to control was previously reported in dairy goat using 6 replications in each group (Total n=12; Nguyen et al., 2018). In the present manuscript, we could see the effect of DCAD on blood pH on the 4th week of supplement. This observation suggested that we could demonstrate the effect of DCAD with this number of replication. In terms of AA, this supplement protocol did decrease plasma MDA in this study. Therefore, the current positive results would be an effect of supplements. We agree with the reviewer and thank you very much to remind us this point (very good) that with our experimental design, the interpretation of negative results (physiological responses) should be carefully made. We have discussed this issue and try to compromise in the discussion (L246-248).  

Line 173-175, abbreviations used in Table 2 need to be defined as a footnote as table need to provide adequate information without referring to the other section of the manuscript.

Ans: Thank you, we do appreciate this suggestion and we did provide the abbreviation on L177-178.

L177-178 “…Abbreviations; RR respiration rat; Tr, rectal temperature;.. “

It is generally well described. The main weakness or limitation of this study is that the number of replications utilized is too low (3 replication per treatment).

Ans: We already added more information as previously described in the first question.

Discussion

Line 294-295, In a statement “Whether these were the direct AA effect of mammary gland oxidative status” change direct AA effect ‘of’ to direct AA effect ‘on’.

Ans: Thank you for suggestion and we did change this word on L300.

L300 “…. to direct AA effect on mammary gland…”

Conclusion

First part is acceptable but the later part seems speculative.

Ans: Thank you, we do appreciate this suggestion. We agree to change this conclusion part.

L342   “……oxidative stress might involve in….”
